# Design for Manufacturing and Assembly (DfMA) Checklists for Off-Site Construction (OSC) Projects

Seoyoung Jung  and Jungho Yu *

Department of Architectural Engineering, Kwangwoon University, Seoul 01897, Korea
* Correspondence: myazure@kw.ac.kr; Tel.: +82-2-940-5564

**Abstract:** Off-Site Construction (OSC), which has the advantage of improving construction productivity, is being spotlighted as a solution to the limitations of conventional construction production methods. Despite the need for, and various advantages of, the introduction and utilization of OSC, however, several issues remain, such as design errors and reduction in design completeness, due to the lack of experience and expertise of project participants, as well as improper consideration of production environment and technical constraints. To resolve these issues, it is necessary to develop an optimal design plan that conforms to the OSC manufacturing environment and manufacturing efficiency; thus, there have been ongoing efforts in the construction industry to introduce Design for Manufacturing and Assembly (DfMA), to derive the optimal design plans for OSC projects. Some studies related to the application of DfMA to OSC have been conducted, however they neglected to present a checklist for reviewing the optimality of OSC design plans. This study has therefore developed an OSC–DfMA checklist, to review the optimality of design plans for OSC projects, by listing optimal design goals for OSC projects, the OSC process, and DfMA principles. This study utilized the systematic literature review, structured interview, and content validity analysis methods to develop the OSC–DfMA checklist presented herein. The developed OSC–DfMA checklist will be applicable to reviewing the optimality of the OSC design plans.

**Keywords:** Off-Site Construction; Design for Manufacturing and Assembly; precast concrete; optimum design; design review

## 1. Introduction

Off-Site Construction (OSC) boasts enhanced productivity through standardization, modularization, and repeated production, as well as manufacturing quality and safety, due to the reduction in outdoor work. These advantages have been spotlighted as a solution to the limitations of conventional construction methods. Despite the need, however, for the introduction and utilization of OSC, and despite its various strengths, the OSC method has yet to be widely accepted in the construction industry, because of design errors and low completeness [1,2], due to manufacturing environment and technical restraints [3–5], as well as insufficient experience and project participants' lack of expertise [1,3,6–9]. To overcome these issues, optimal design with high manufacturing efficiency, tailored to the manufacturing environment and techniques—including factory facilities, transport equipment, lifting equipment, and the assembly method—is urgently needed [10,11].

The construction industry is being heedful of the Design for Manufacturing and Assembly (DfMA), to derive the optimal design plan for OSC projects. DfMA is a concept that has been developed to minimize design changes in the manufacturing industry. DfMA is a design approach that prevents possible errors, in advance, in the production and assembly phases, and improves production efficiency by inspecting various circumstances related to the product production phase in advance of the design phase. The OSC leaders, including Singapore, the United Kingdom, and the United States, are providing DfMA guidelines suitable for the characteristics of the OSC industry, by applying the concept of

DfMA. However, most of the DfMA guidelines, that have thus far been developed in the OSC field, aim to provide background knowledge for the OSC manufacturing process, thus showing limitations in evaluating the optimality of design plans. Many researchers have therefore been developing new methods for evaluating the optimality of OSC project-design plans. These studies, however, have only highlighted the aspect of manufacturing efficiency, neglecting the consideration of its suitability for the OSC manufacturing environment.

In this respect, this study aimed to develop a DfMA checklist, from the dimension of conformity of production and manufacturing efficiency to the OSC manufacturing environment (hereinafter referred to as 'OSC–DfMA checklist'), to support the selection of the optimal design plan for OSC. To achieve this research goal, this study first conducted a systematic literature review to derive preliminary DfMA items from the DfMA literature in the conventional OSC field, and then conducted structured interviews with experts, related to OSC, to derive the secondary DfMA items. Finally, this study utilized the content validity analysis method, to present the final OSC–DfMA checklist.

## 2. Literature Review

### 2.1. Optimal Design for OSC Projects

The OSC method, unlike conventional manufacturing methods, has many restrictions in manufacturing environment and techniques, including factory facilities, transport equipment, lifting equipment, and assembly method [3–5]; the OSC method therefore requires the development and selection of design plans with the highest manufacturing efficiency among the design plans, in consideration of these restrictions [12]. When these restrictions are improperly considered during the design process, or when the optimal alternative in terms of manufacturing efficiency is not selected, a delayed delivery due to redesign or rework may result in negative outcomes, such as reduction in productivity, work safety, and manufacturing quality.

Thus, Sharafi et al. (2017) have claimed that there is a need for design that considers a proper balance between the goals and constraints of a project, as well as the basic performance of a building, and that, to this end, considerations must include injection costs, structural safety, and the energy-efficiency of the project [10]. Furthermore, Salama et al. (2017) have presented a method of optimizing the configuration of a module, by utilizing indicators such as the distance to the construction site, the size of the module, the number of junction points, and operating conditions and related costs for cranes, as well as basic measurements according to the volume of concrete, to optimize the design of OSC buildings while considering the restrictions of OSC [11]. Moreover, Smith (2010) has emphasized the consideration of the optimal configuration of a module, including transport, assembly, crane, and restriction in tolerance in the modular design process [13].

By contrast, Gbadamosi et al. (2019) have emphasized the need to introduce the concepts of DfMA and Lean Construction, in order to realize the optimal design for OSC projects; accordingly, they presented a method of evaluating a design alternative in terms of: (1) ease of assembly; (2) ease of handling; (3) assembly speed; and (4) assembly waste [12]. The same study emphasized the selection of a design plan with high manufacturing efficiency among multiple design alternatives, to achieve optimal design. In addition, Said et al. (2017) have developed an optimal model for resolving the balance between manufacturing efficiency and design flexibility, emphasizing that the OSC must secure manufacturing efficiency through the standardization of constituent components, as well as diversity in construction design [14].

In this manner, previous studies regarding the optimal design of OSC have suggested that the suitability of an OSC manufacturing environment must be considered through a design plan that reflects matters related to manufacturability, safety, and quality in the manufacturing environment, such as transport and lifting conditions during the design stage of OSC; furthermore, these studies suggest that the design plan must reflect items directly related to the efficiency of manufacturing, such as the size of a module, the number of junction points, and the utilization of devices to select the optimal alternative. This study

has thus proceeded by defining the optimal design for OSC projects as a design with high manufacturing efficiency, which enables production in terms of manufacturability, manufacturing safety, and manufacturing quality, by reflecting considerations related to the entire manufacturing process (manufacturing, transport, on-site assembly, operation and maintenance) in a given manufacturing environment (including a factory and site facilities).

### 2.2. Design for Manufacturing and Assembly (DfMA)

Traditionally, designers in the manufacturing field have regarded design and production as separate, based on the attitude that "we design it; you build it!" or an 'over the wall' approach. Such an approach, however, requires the undertaking of redesign work, in the event of design-modification issues at the manufacturing or assembly stage, after the completion of the design; in such cases, design modification is very costly. The DfMA, which was formulated to remove this limitation, refers to a design approach that seeks to prevent errors that might occur at the manufacturing and assembling stages, by examining, at the design stage, various situations related to the product manufacturing stage.

As such, various DfMA principles have been developed from various viewpoints, including cost reduction, production period reduction, and the implementation of environmentally friendly designs, as shown in Table 1 [15–20]), and by various researchers, including Bogue (2012), Emmatty and Sarmah (2012), Krumenauer et al. (2008), Stoll (1986), and Boothroyd (1994) and research institutes, to reduce difficulties during the process of manufacturing and assembling products by a designer. Although the principles presented by the researchers have varied slightly, depending on which viewpoint they were more concerned about, most of them are similar in that they focus on minimizing the number of parts, simplifying assembly and handling, standardizing, and minimizing waste.

**Table 1.** General DfMA principles.

| No. | DfMA Principles | Source |
|---|---|---|
| 1 | It is necessary to prevent unnecessary reworking through error-free design, and to ensure safety and quality during the manufacturing process. | [15–19] |
| 2 | It is necessary to minimize waste, through design that considers re-usability. | [17] |
| 3 | It is necessary to prevent errors during the process of product handling and assembly, through design that considers production environment and process. | [15,17] |
| 4 | It is necessary to reduce manufacturing time and cost, by avoiding complex assembly methods, through design that considers ease of manufacturing. | [15,17–20] |
| 5 | It is necessary to reduce the time and cost consumed for the handling process, by simplifying the method of handling and assembling parts. | [15,17–20] |
| 6 | It is necessary to reduce the time and cost consumed for the assembly process, by performing design based on the assembly method. | [15,17,19,20] |
| 7 | It is necessary to simplify design through modular design. | [15,17–19] |
| 8 | It is necessary to minimize manual labor, and secure product quality and assemblyefficiency, through a design that applies a mechanical assembly method. | [15–17,19,20] |
| 9 | It is necessary to reduce the time and cost consumed for purchasing parts, through design that repeatedly uses standardized parts. | [15,16,18–20] |
| 10 | It is necessary to simplify the manufacturing process, by repeatedly using similar materials. | [15,19,20] |
| 11 | It is necessary to minimize the impact on the environment, through the selection of environmentally friendly materials and the minimization of waste. | [16,17,20] |
| 12 | It is necessary to reduce the time and cost consumed for the manufacturing and assembly process, through design with a minimum number of parts. | [15–20] |
| 13 | It is necessary to reduce the time and cost consumed for the manufacturing process, by standardizing connector types and minimizing the number of connectors. | [15–20] |
| 14 | It is necessary to reduce component failure, by minimizing the use of fragile components. | [15,19,20] |
| 15 | It is necessary to reduce time and money consumed for the manufacturing process, by minimizing finishing work. | [15,17,19,20] |

There are various other DfMA principles, and numerous companies in the manufacturing field have experienced the positive effects—including reduction in the product development period, enhanced productivity, improved product quality, enhanced reliability in design, reduction in waste resources, and improved profitability—of adhering to these principles throughout the entire design process.

The DfMA has recently expanded from Design for Manufacturing and Assembly to 'Design for Quality'; 'Design for Safety'; 'Design for Service' (which considers after-safe services), and 'Design for Environment', (which considers environmental impact). The DfMA is alternatively called 'Design for Excellence (DFX)', referring to a design for all fields.

*2.3. Current Status of DfMA Guidelines in the Construction Field, by Country*

The OSC leaders, which include Singapore, the United Kingdom, and the United States, are providing DfMA guidelines and standards that apply DfMA under the government's lead, for the optimal design of OSC projects, to increase the productivity of OSC projects. Representative DfMA development examples are illustrated below.

(1)    Singapore

Singapore, under the lead of the Building and Construction Authority (BCA), defines DfMA as various technologies and methodologies that can promote OSC and improve the productivity of construction. The BCA selected DfMA as one of the core technologies for innovations in the construction industry, in the Construction Industry Transformation Map, 2017. The BCA provides DfMA guidelines in six sectors, ranging from individual material units to complete assemblies (advanced precast concrete systems; mass-engineered timber; prefabricated, prefinished volumetric construction; prefabricated bathroom units; prefabricated mechanical, electrical, and plumbing systems; and structural steel), with the core goals of simplification of on-site assembly, improvement of on-site productivity, reduction of input labor, and improvement of quality and safety [21]. These guidelines describe the overall requirements for achieving the core goals, ranging from project-planning to matters regarding design, transport, installation, quality inspection, maintenance and repair, and the related systems for presenting the details to be considered, in designing and manufacturing activities by project type.

(2)    The United Kingdom

The U.K. Royal Institute of British Architects (RIBA), starting from the RIBA Plan of Work 2013: Design for Manufacture and Assembly [22], has underlined the need for the application of DfMA, noting that it will engender positive effects, including: the shortening of construction periods by 20–60%; the reduction of construction costs by 30–40%; the reduction of on-site labor by 70% or more; quality improvement; safety assurance; and reduced construction waste.

Subsequently, the RIBA published the first edition of the DfMA Overlay to the RIBA Plan of Work in 2016, and then its revised edition in 2021 [23], to present DfMA application strategies for each step of the manufacturing process. The RIBA's manufacturing process includes the following eight steps: strategic definition; preparation and briefing; concept design; spatial coordination; technical design; manufacturing and construction; handover; and use.

Furthermore, this document insists that successful project management reduces project risks during the manufacturing and construction process—through initial collaboration among design companies, manufacturing and supplying companies, and construction companies—and presents the following design considerations for a successful DfMA process:

-    consideration of connectivity between OSC components;
-    provision of appropriate tolerances, to ensure ease of manufacturing and assembly;
-    repeated use of standardized components;

- optimization of module configuration, considering the functions and optimal bonding methods of OSC components;
- preliminary reflection of issues that occur during maintenance, repair, and dismantling.

(3)    The United States

The Modular Building Institute (MBI), an international non-profit trade association, presented the ICC/MBI 1200-2021 [24]—a standard regarding planning, design, fabrication, and assembly for OSC—in cooperation with the International Code Council (ICC), in 2019. This standard—despite not explicitly mentioning DfMA—can be applied as a guideline for implementing DfMA-based optimal design, because it prescribes the roles of designers, module manufacturers, construction mangers, and construction companies, and because it describes key management requirements ranging from architectural and structural design, through manufacturing, transport, and storage of modules, to on-site installation.

Moreover, the American Institute of Architects (AIA) released 'Design for Modular Construction: An Introduction for Architects', in collaboration with the National Institute of Building Sciences (NIBS), in 2019 [25]. This guide presents the advantages of modular design, by exemplifying successful design cases of modular construction. Furthermore, this guide emphasizes that the integration of design, manufacturing, transport, and on-site assembly stages is essential in the OSC manufacturing method, and that DfMA should be introduced for the integration of the entire production process. From this viewpoint, designers are urged to create a design based on understanding of the process, covering manufacturing modules at the factory, transport, and on-site assembly levels.

(4)    China

The Ministry of Housing and Urban–Rural Development announced the Standard for Design of Assembled Housing in 2019 [26]. This design standard—despite not directly mentioning DfMA—seeks standardization, modularization, and miniaturization corresponding to the basic DfMA principles, and presents design considerations that meet the requirements of the entire production method, covering everything from standardized design, through factory manufacturing, to on-site installation.

*2.4. Previous Studies Related to DfMA in the Construction Field*

There is recognition of the need to consider construction, maintenance, and repair processes at the design stage, by applying the concept of DfMA in the construction industry. Several researchers have applied DfMA to construction projects, to verify its effects—such as enhanced productivity, improved quality, and enhanced safety. Banks et al. (2018) [27] have analyzed successful DfMA application cases in the U.K., from the aspects of structural design, production, safety, and assembly, to reveal DfMA's effects—such as improved construction speed, reduced cost, enhanced work safety, improved quality, sustainability through reduced waste, and secured reliability of projects. Moreover, Chen and Lu (2018) [28] have conducted a literature review, and have identified the reduced number of parts and fixture parts for curtain walls, the use of feasible materials, the composition of size and weight for parts, to ensure ease of handling, and the reduced waste of materials, based on DfMA principles applicable to curtain wall systems (CWS); they thus verified the advantages of reduced material cost, shortened on-site assembly time, reduced waste, and improved quality.

Many studies have proved the application of DfMA to be effective in enhancing productivity and quality in construction projects, and have presented directions for applying DfMA to construction projects [2,12,29–32]. Kim et al. (2016) [29] presented DfMA-based design criteria for bridge construction, which were derived from the viewpoint of four representative design criteria for DfMA in the manufacturing field (simplification of design; minimization of parts; standardization of parts and materials; and simplification of parts-handling and assembly work), to improve its efficiency. Furthermore, Tan et al. (2020) [30] presented five development directions for DfMA guidelines applicable to the construction industry (context-based design; technology-rationalized design; logistics-optimized de-

sign; component-integrated design; and material-lightened design), in consideration of the differences between manufacturing and construction industries, by analyzing existing DfMA guidelines. Similarly, Lu et al. (2021) [31] presented the directionality of DfMA in the construction industry, by analyzing and comparing the current development status of DfMA between the manufacturing and construction industries, as well as similar concepts to DfMA. The same study suggested that DfMA is necessary for the development of construction materials and production and assembly technologies, as well as continuous reinforcement of logistics and the supply chain. In contrast, Hyun et al. (2022) [2] pinpointed design errors—owing to omission of considerations regarding OSC projects, due to participants' lack of experience and knowledge—as a delay factor in OSC projects, and emphasized that DfMA should be applied to resolve this issue; they proposed an OSC design process, to which the DfMA principles were applied, and verified the research results through case studies.

These previous studies are significant, in that they present development directions for the application of DfMA, from the manufacturing field to the construction field. These studies' results cannot be utilized, however, in reviewing the optimality of design plans in the OSC design practice, because they provide no detailed review items for DfMA.

Meanwhile, several studies have been conducted to evaluate the optimality of OSC design plans. Gbadamosi et al. (2019) [12] identified DfMA, Lean Construction principles, and optimization factors (including the weight of components, the number of on-site workers, and the number of parts), and developed a method for evaluating design alternatives—consisting of four detailed indicators (ease of assembly, ease of handling, assembly waste, and assembly speed)—by utilizing them. Moreover, Safaa et al. (2019) [32] have presented a DfMA evaluation method that can assess the DfMA satisfaction levels for each precast concrete (PC) component, and which is applicable to Accelerated Bridge Construction (ABC), a method for building a bridge by utilizing pre-assembly techniques. The same study utilized indicators for evaluating design alternatives—the degree of standardization, the number of components, the simplicity of design, and the ease of handling—to present an evaluation method for selecting alternatives with high manufacturing efficiency.

These studies are significant, in that they identify the evaluation factors for DfMA in relation to manufacturing efficiency, such as the number of components, and the assembly time. However, these studies emphasize only the manufacturing efficiency aspect, during the process of identifying DfMA items, and neglect consideration of production availability, production quality, and production safety depending on constraints under the OSC environment.

During the selection of design alternatives with high production efficiency, for optimal design to minimize the design modification by minimizing design errors for OSC, the question is prioritized during the design process, of whether a design plan is suitable for the OSC manufacturing environment. Thus, this study aimed to divide DfMA evaluation items into aspects of conformity of production suitability (production availability, production quality, and production safety) and production efficiency, and to propose a method for evaluating the conformity of production during the process of creating design alternatives, and a method for evaluating manufacturing efficiency, that could be utilized during the process of selecting design alternatives.

## 3. Scope and Method

This study aimed to develop a DfMA checklist, to support decision-making in the development and selection process of optimal design plans for OSC projects. OSC projects can be classified into various types, depending on the size and shape of basic units constituting a building, applied materials, and the degree of pre-assembly. This study focused on the OSC with the most fundamental precast concrete PC method among various OSC types.

To achieve its research goals, this study developed an OSC–DfMA checklist, as shown in Figure 1, by combining the systematic literature review, structured interview, and content validity analysis methods. Firstly, to derive items based on clear criteria, this

study established a frame in which to derive a primary set of preliminary OSC–DfMA items reflecting the goals of optimal OSC design, the OSC production process, and DfMA principles; the study then utilized this frame, to examine the literature related to the existing construction field, and to derive a primary set of preliminary OSC–DfMA items. Next, interviews were conducted with experts who had experience of participating in OSC projects in South Korea, and the results of the interview were used to derive a secondary set of preliminary OSC–DfMA items.

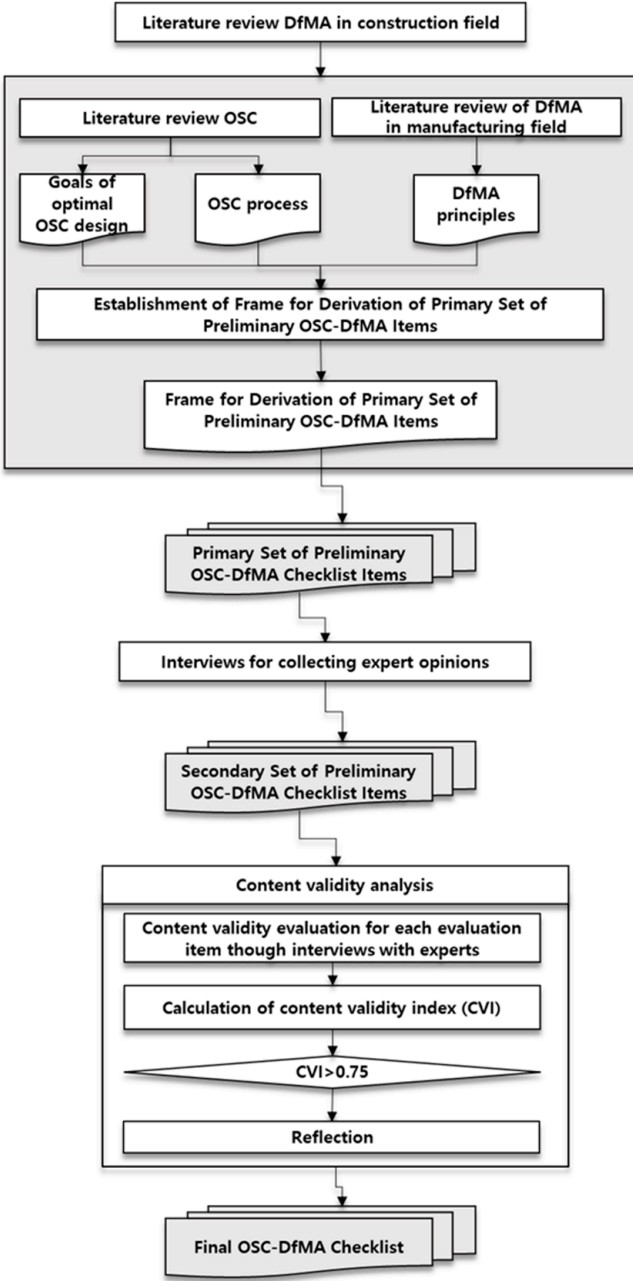

**Figure 1.** The method for deriving OSC–DfMA items.

Finally, the content validity of each preliminary item was evaluated, by conducting interviews with South Korean OSC experts who had not participated in the previous interview. The content validity—also known as definition validity and logical validity—can be defined as the ability of the selected items to reflect the variables of the construct in the measure [33]. In this study, the experts were asked to evaluate the content validity of each item on a 4-point scale, and the content validity index (CVI) of each item was calculated,

using the results of the evaluation by the experts. In this instance, the method of calculating the content validity index was that used by Lynn (1986) [34]; only items with a content validity index exceeding 0.75 were selected as the final items.

## 4. Development of the OSC–DfMA Checklist

### *4.1. Derivation of the Primary Set of Preliminary OSC–DfMA Items*

4.1.1. Establishment of the Frame for Derivation of the Primary Set of Preliminary OSC–DfMA Items

This study established a frame for derivation of a primary set of preliminary OSC–DfMA items, from which to derive OSC–DfMA evaluation items, based on a specific process reflecting the requirements for optimal OSC design and DfMA principles in a process of OSC.

The optimal design of OSC is defined as a design with high manufacturing efficiency, that enables production in terms of manufacturability, manufacturing safety, and manufacturing quality, by reflecting considerations related to the entire manufacturing process of OSC (factory manufacturing, transport, on-site assembly, and operation and maintenance) in a given production environment (including factory and site facilities). Thus, the optimal design of OSC has four goals: production availability; production safety; production quality; and production efficiency. In addition, as discussed in Section 2.2, this study defined DfMA in the manufacturing field by nine principles (consideration of manufacturing conditions; consideration of risk factors; minimization of parts breakage; securement of the quality of junction parts; minimization of the number of parts; standardization of parts; consideration of the reusability of parts; minimization of additional work; and simplification of assembly and handling methods) by considering the industrial characteristics of OSC.

The results of applying the nine DfMA principles to the requirements for optimal OSC design are presented as follows. The design had to consider the manufacturing conditions, to secure manufacturability, and the risk factors in the manufacturing process had to be considered, to secure production safety. To secure manufacturing quality, the design had to ensure the minimization of parts breakage, and the securement of the quality of junction parts. Furthermore, to improve manufacturing efficiency, the design had to consider minimization of the number of parts, standardization of parts, consideration of the reusability of parts, minimization of additional work, and simplification of assembly and handling methods.

In this manner, this study established a Frame for Derivation of a primary set of preliminary OSC–DfMA items as shown in Figure 2, by reflecting the OSC manufacturing process (manufacturing stage; transport stage; on-site assembly stage; and operation and maintenance stage) in the requirements for optimal OSC design and DfMA principles.

4.1.2. Derivation of Primary Set of Preliminary OSC–DfMA Items

Next, this study derived a primary set of preliminary OSC–DfMA items, by systematically analyzing and itemizing the DfMA guidelines and previous studies related to DfMA in the OSC field in Singapore, the United Kingdom, and the United States, utilizing the previously established Frame for Derivation of OSC–DfMA evaluation items. To derive a primary set of preliminary items, this study utilized the DfMA guidelines by BCA (2018) [21], RIBA (2020) [23], MBI (2019) [24], and China's Ministry of Housing and Urban–Rural Development [26]. The study also used research by Tan et al. (2020) [30], Gbadamosi et al. (2019) [12], Safaa et al. (2019) [32], and Kim et al. (2016) [29].

The items identified as a primary set of preliminary items are shown in Table 2. As a result of the identification, 16 primary preliminary items related to the manufacturing stage were identified, and 9 primary preliminary items related to the transport stage were identified. In addition, 19 primary preliminary items related to the on-site assembly stage were identified, and 5 items related to the operation and maintenance stage were identified.

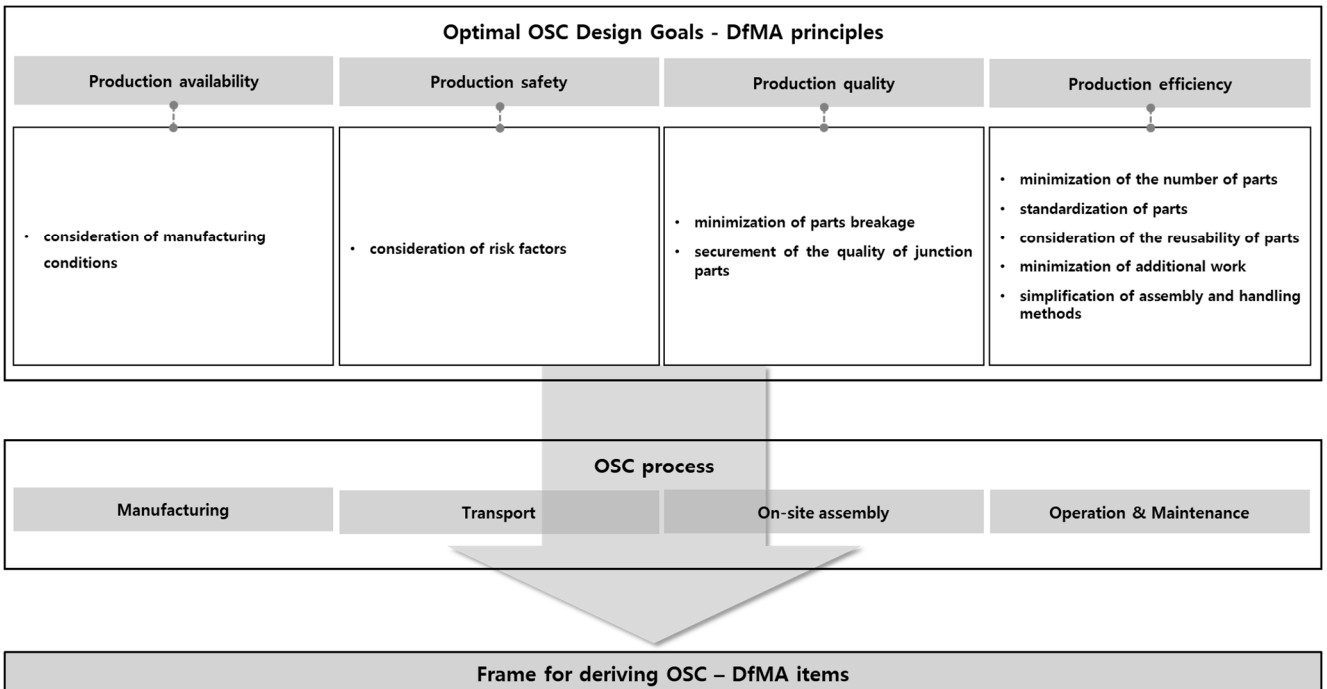

**Figure 2.** Overview of the Frame for Derivation of a primary set of preliminary OSC–DfMA items.

**Table 2.** The primary set of preliminary items.

| OSC Stage | Goals of Optimal OSC Design | DfMA Principles | The Primary Set of Preliminary Items for DfMA | Source |
|---|---|---|---|---|
| manufacturing stage | Production availability | Consideration of manufacturing conditions | • Have you reviewed the size/weight/configuration of the module in consideration of the factory's lifting equipment? | [21,23,24,26,30] |
| | Production safety | Consideration of risk factors | • Have you reviewed possible safety issues in the production process of the factory? | [21,23,24] |
| | | | • Have you reviewed the location of the lifting point and the balance of the module during lifting? | [21,23,24] |
| | Production quality | Minimization of parts breakage | • Have you reviewed the curing measures for the PC module? | [21,23,24] |
| | | | • Have you reviewed the structural performance (including stress and deformation) of the lifting point designated for each module? | [21,24] |
| | | | • Have you provided the loading conditions (type, position, and spacing of the pedestal)? | [21,24] |
| | | | • Have you reviewed the structural performance of the fixing device (anchor bolt) installed at the lifting point? | [21,24] |

**Table 2.** *Cont.*

| OSC Stage | Goals of Optimal OSC Design | DfMA Principles | The Primary Set of Preliminary Items for DfMA | Source |
|---|---|---|---|---|
| | | Securement of the quality of junction parts | • Have you reviewed the performance (position and route) of the piping connection for water supply, drainage, and sanitation facilities? | [21,24] |
| | | | • Have you designed considering the application of an appropriate ventilation system? | [21] |
| | | | • Have you reviewed the performance of plumbing connections for electrical and telecommunication facilities? | [21,24] |
| | Production efficiency | Minimization of the number of parts | • Have you reviewed the possibility of minimizing the number of modules? | [12,21,23,24,29,32] |
| | | Standardization of parts | • Have you minimized the number of mould types? | [12,21,23,24,29,32] |
| | | | • Have you minimized the number of module types? | [12,21,23,24,29,32] |
| | | Consideration of the reusability of parts | • Have you designed considering the reusability of the mould? | [12,21,29] |
| | | Minimization of additional work | • Not applicable | |
| | | Simplification of assembly and handling methods | • Have you properly configured the size and weight of the module? | [12,21,23,30,32] |
| | | | • Have you designed considering the difficulty of manufacturing the module? | [12,21,23,29,32] |
| Transport stage | Production availability | Consideration of manufacturing conditions | • Have you reviewed the size/weight/configuration of the module, in consideration of transportation equipment? | [21,23,24,26,30] |
| | | | • Have you reviewed the size/weight/configuration of the module, in consideration of road conditions inside and outside the site? | [21,24,26,30] |
| | Production safety | Consideration of risk factors | • Have you reviewed possible safety issues during the transport process, and presented a transport method that can ensure stable transport? | [21,26] |
| | Production quality | Minimization of parts breakage | • Have you reviewed the possibility of module deformation or cracks during transport? | [21,24] |
| | | Securement of the quality of junction parts | • Not applicable | - |

**Table 2.** *Cont.*

| OSC Stage | Goals of Optimal OSC Design | DfMA Principles | The Primary Set of Preliminary Items for DfMA | Source |
|---|---|---|---|---|
| | Production efficiency | Minimization of the number of parts | • Have you reviewed the possibility of minimizing the number of modules? | [21,23,26,29,32] |
| | | Standardization of parts | • Have you minimized the number of module types? | [12,21,23,26,29,32] |
| | | Consideration of the reusability of parts | • Not applicable | - |
| | | Minimization of additional work | • Have you minimized the use of special transport equipment? | [12,21,32] |
| | | Simplification of assembly and handling methods | • Have you properly configured the size and weight of the module ? | [12,21,23,29,32] |
| | | | • Have you minimized the number of types of transport equipment required to transport from the factory to the site? | [12,21,29,32] |
| on-site assembly stage | Production availability | Consideration of manufacturing conditions | • Have you considered the size/weight/configuration of the module in consideration of the lifting equipment on the site? | [21,24,26,30] |
| | | | • Have you reviewed the size/weight/configuration of the module in consideration of the field layout of the lifting equipment? | [21,24,30] |
| | | | • Have you properly planned the construction work of the junction joining method between modules? | [21,24,26,30] |
| | | | • Have you simulated the configuration of all junctions between modules in advance? | [21,23,30] |
| | Production safety | Consideration of risk factors | • Have you presented an open-storage method by reviewing possible safety issues during open-storage work? | [21,23,26] |
| | | | • Have you established a lifting plan by reviewing possible safety issues during lifting? | [21,23,26] |
| | | | • Have you established a joining plan by examining possible safety issues during the joining process? | [21,23,24,26] |

**Table 2.** *Cont.*

| OSC Stage | Goals of Optimal OSC Design | DfMA Principles | The Primary Set of Preliminary Items for DfMA | Source |
|---|---|---|---|---|
| | Production quality | Minimization of parts breakage | • Not applicable | |
| | | Securement of the quality of junction parts | • Have you selected the joining method in consideration of the junction spacing and the level of stress? | [21] |
| | | | • Have you reviewed the structural performance of the module junction? | [21,24,26] |
| | | | • Have you reviewed the use performance (including watertightness, fire resistance, durability, insulation, and sound insulation) of the module junction? | [21,26] |
| | | | • Have you reviewed the ease of vertical/horizontal adjustment of the joining method between modules? (Is it easy to adjust?) | [21,24] |
| | | | • Have you applied a joining material (such as grouting material or hardware) that can ensure good performance against chemical and physical influences? | [21,24] |
| | Production efficiency | Minimization of the number of parts | • Have you reviewed the possibility of minimizing the number of modules? | [12,21,24,26,29,32] |
| | | Standardization of parts | • Have you minimized the number of module types? | [12,21,24,26,29,32] |
| | | | • Have you minimized the number of types of joining methods? | [21,24,32] |
| | | Consideration of the reusability of parts | • Not applicable | |
| | | Minimization of additional work | • Have you minimized curing management and special finishing work at the junction point? | [12,21] |
| | | Simplification of assembly and handling methods | • Have you properly configured the size and weight of the module? | [12,21,24,26,29,32] |
| | | | • Have you minimized the number of junction points of the module? | [12,21,24,26,29,32] |
| | | | • Have you reviewed the difficulty of the joining method? | [12,21,24,29,32] |

**Table 2.** *Cont.*

| OSC Stage | Goals of Optimal OSC Design | DfMA Principles | The Primary Set of Preliminary Items for DfMA | Source |
|---|---|---|---|---|
| Operation and maintenance stage | Production availability | Consideration of manufacturing conditions | • Have you reviewed the location of periodic maintenance activities (including electricity, firefighting, gas, water supply, and rescue) conducted during the building use phase to minimize user inconvenience? | [21] |
| | Production safety | Consideration of risk factors | • Not applicable | - |
| | Production quality | Minimization of parts breakage | • Not applicable | - |
| | | Securement of the quality of junction parts | • Have you checked and repaired defects (including cracks, leaks) of the junction during the building use phase? | [21,23] |
| | Production efficiency | Minimization of the number of parts | • Not applicable | - |
| | | Standardization of parts | • Not applicable | - |
| | | Consideration of the reusability of parts | • Not applicable | - |
| | | Minimization of additional work | • Not applicable | - |
| | | Simplification of assembly and handling methods | • Is it easy to observe whether a module may have defects, and to access it for maintenance and repair? | [21] |
| | | | • Have you reviewed the difficulty of maintenance and repair of the junction? | [21,23] |
| | | | • Have you reviewed the ease of detachment of non-structural partition walls, to improve the possibility of remodeling? | [21] |

*4.2. Derivation of the Secondary Set of Preliminary OSC–DfMA Items*

This study further conducted structured interviews with experts who had experience of participating in PC-based OSC projects, to collect their opinions. The expert interviews were conducted via email from 15 October to 15 November 2020, targeting 11 relevant experts. Each expert interview was conducted by an expert reading the first preliminary items and then responding to the items to be modified, deleted, and added. The demographic characteristics of the experts that participated are described in Table 3.

Table 4 presents a secondary set of preliminary items for the manufacturing stage, which were derived from the opinions of experts. All the experts who participated in the interview agreed on most of the primary preliminary items related to this stage. In addition, PC-manufacturing and structural-design experts suggested that factory manufacturing tolerances should be presented in the design stage, and that the possibility of module deformation, cracks, and partial damage should be considered in the factory manufacturing process. Therefore, in this study, 18 secondary preliminary items were identified by adding such expert opinions.

**Table 3.** Demographic characteristics of experts that participated in interviews.

| | Category | Frequency | % |
|---|---|---|---|
| | Ordering Organization | 2 | 18.18% |
| | Architectural Design | 2 | 18.18% |
| Organization type | Structural Design | 2 | 18.18% |
| | PC Manufacturing | 1 | 9.09% |
| | Construction | 1 | 9.09% |
| | Academia | 1 | 9.09% |
| Years of experience | Construction-related | 19.8 years | |
| | OSC-related | 4.2 years (3.1 times) | |
| | Total | 11 | 100% |

**Table 4.** The secondary set of preliminary items for the manufacturing stage.

| Goals of Optimal OSC Design | DfMA Principles | Primary Set of Preliminary Items | Expert Opinions | Secondary Set of Preliminary Items |
|---|---|---|---|---|
| Production availability | Consideration of manufacturing conditions | • Have you reviewed the size/weight/configuration of the module in consideration of the lifting equipment at the factory? | • (Modification) In addition to the lifting equipment at the factory, it is necessary to consider various factory manufacturing facilities.<br>• (Modification) It is also necessary to consider the shape of the module whose mould can be removed. | • Have you reviewed the size/weight/shape/configuration of the module in consideration of the manufacturing facilities (such as lifting equipment) at the factory? |
| | | - | • (Addition) It is necessary to present the tolerance for factory manufacturing. | • Was the tolerance for factory production presented ? |
| Production safety | Consideration of risk factors | • Have you reviewed possible safety issues in the production process of the factory? | • Agreed | • Have you reviewed possible safety issues in the production process of the factory? |
| | | • Have you reviewed the location of the lifting point and the balance of the module during lifting? | • Agreed | • Have you reviewed the location of the lifting point and the balance of the module during lifting? |
| Production quality | Minimization of parts breakage | • Have you reviewed the curing measures for the PC module? | • Agreed | • Have you reviewed the curing measures for the PC module? |
| | | • Have you reviewed the structural performance (including stress and deformation) of the lifting point designated for each module? | • Agreed | • Have you reviewed the structural performance (including stress and deformation) of the lifting point designated for each module? |
| | | • Have you provided the loading conditions (type, position, and spacing of the pedestal)? | • Agreed | • Have you provided the loading conditions (type, position, and spacing of the pedestal)? |
| | | • Have you reviewed the structural performance of the fixing device (anchor bolt) installed at the lifting point? | • Agreed | • Have you reviewed the structural performance of the fixing device (anchor bolt) installed at the lifting point? |
| | | - | • (Addition) Have you reviewed the possibility of module deformation, cracks, and partial breakage during the factory manufacturing process? | • Have you reviewed the possibility of module deformation, cracks, and partial breakage during the factory manufacturing process? |

**Table 4.** *Cont.*

| Goals of Optimal OSC Design | DfMA Principles | Primary Set of Preliminary Items | Expert Opinions | Secondary Set of Preliminary Items |
|---|---|---|---|---|
| Production quality | Securement of the quality of junction parts | • Have you reviewed the performance (position and route) of the piping connection for water supply, drainage, and sanitation facilities? | • Agreed | • Have you reviewed the performance (position and route) of the piping connection for water supply, drainage, and sanitation facilities? |
| | | • Have you designed considering the application of an appropriate ventilation system? | • Agreed | • Have you designed considering the application of an appropriate ventilation system? |
| | | • Have you reviewed the performance of plumbing connections for electrical and telecommunication facilities? | • Agreed | • Have you reviewed the performance of plumbing connections for electrical and telecommunication facilities? |
| Production efficiency | Minimization of the number of parts | • Have you reviewed the possibility of minimizing the number of modules? | • Agreed | • Have you reviewed the possibility of minimizing the number of modules? |
| | Standardization of parts | • Have you minimized the number of mould types? | • Agreed | • Have you minimized the number of mould types? |
| | | • Have you minimized the number of module types? | • Agreed | • Have you minimized the number of module types? |
| | Consideration of the reusability of parts | • Have you designed considering the reusability of the mould? | • Agreed | • Have you designed considering the reusability of the mould? |
| | Minimization of additional work | • Not applicable | • No further comments | • Not applicable |
| | Simplification of assembly and handling methods | • Have you properly configured the size and weight of the module? | • Agreed | • Have you properly configured the size and weight of the module? |
| | | • Have you designed considering the difficulty of manufacturing the module? | • Agreed | • Have you designed considering the difficulty of manufacturing the module? |

Table 5 presents a secondary set of preliminary items for the transport stage, which were derived from the opinions of experts. All the experts who participated in the interview agreed on most of the primary preliminary items related to this stage. In addition, PC manufacturing and structural design experts suggested adding an item, that the size of the module should be reviewed in consideration of the Road Traffic Act in the design stage. Therefore, in this study, 10 secondary preliminary items were identified by adding the opinions of these experts.

Table 6 presents a secondary set of preliminary items for the on-site assembly stage, which were derived from the opinions of experts. All the experts who participated in the interview agreed on most of the primary preliminary items related to this stage. In addition, experts in the field of structural design emphasized that it is necessary to consider the construction error of the joint during the design stage, and to consider the possibility of access of the work force to the joint point, and suggested adding related items. In addition, experts in the construction field suggested adding items related to this, because it is necessary to review the possibility of deformation, cracks, and partial damage of the module. Therefore, in this study, 25 secondary preliminary items were derived by adding the opinions of these experts.

Table 7 presents the secondary set of preliminary items for the operation and maintenance stage, which were derived from the opinions of experts. All the experts who participated in the interview agreed on most of the primary preliminary items related to this stage. In addition, PC manufacturing experts emphasized the need to consider

increasing the durability of modules and joints, and suggested adding related items. Therefore, in this study, six secondary preliminary items were identified by adding the opinions of these experts.

**Table 5.** The secondary set of preliminary items for the transport stage.

| Goals of Optimal OSC Design | DfMA Principles | Primary Set of Preliminary Items | Expert Opinions | Secondary Set of Preliminary Items |
|---|---|---|---|---|
| Production availability | Consideration of manufacturing conditions | • Have you reviewed the size/weight/configuration of the module, in consideration of transportation equipment? | • Agreed | • Have you reviewed the size/weight/configuration of the module, in consideration of transportation equipment? |
| | | • Have you reviewed the size/weight/configuration of the module, in consideration of road conditions inside and outside the site? | • Agreed | • Have you reviewed the size/weight/configuration of the module, in consideration of road conditions inside and outside the site? |
| | | - | • (Addition) Have you chosen the size of the module, in consideration of the Road Traffic Act? | • Have you chosen the size/weight/configuration of the module, in consideration of the Road Traffic Act? |
| Production safety | Consideration of risk factors | • Have you reviewed possible safety issues during the transport process, and presented a transport method that can ensure stable transport? | • Agreed | • Have you reviewed possible safety issues during the transport process, and presented a transport method that can ensure stable transport? |
| Production quality | Minimization of parts breakage | • Have you reviewed the possibility of module deformation or cracks during transport? | • (Modification) The most common quality defect during the transport process is partial damage to edges. It is thus necessary to consider partial damage to areas such as edges as well. | • Have you reviewed the possibility of module deformation, cracks, and partial breakage during the transport process? |
| | Securement of the quality of junction parts | • Not applicable | • No further comments | • Not applicable |
| Production efficiency | Minimization of the number of parts | • Have you reviewed the possibility of minimizing the number of modules? | • Agreed | • Have you reviewed the possibility of minimizing the number of modules? |
| | Standardization of parts | • Have you minimized the number of module types? | • Agreed | • Have you minimized the number of module types? |
| | Consideration of the reusability of parts | • Not applicable | • No further comments | • Not applicable |
| | Minimization of additional work | • Have you minimized the use of special transport equipment? | • Agreed | • Have you minimized the use of special transport equipment? |
| | Simplification of assembly and handling methods | • Have you properly configured the size and weight of the module? | • Agreed | • Have you properly configured the size and weight of the module? |
| | | • Have you minimized the number of types of transport equipment required to transport from the factory to the site? | • Agreed | • Have you minimized the number of types of transport equipment required to transport from the factory to the site? |

**Table 6.** The secondary set of preliminary items for the on-site assembly stage.

| Goals of Optimal OSC Design | DfMA Principles | Primary Set of Preliminary Items | Expert Opinions | Secondary Set of Preliminary Items |
|---|---|---|---|---|
| Production availability | Consideration of manufacturing conditions | • Have you considered the size/weight/configuration of the module, in consideration of the lifting equipment on the site? | • Agreed | • Have you considered the size/weight/configuration of the module, in consideration of the lifting equipment on the site? |
| | | • Have you reviewed the size/weight/configuration of the module, in consideration of the field layout of the lifting equipment? | • Agreed | • Have you reviewed the size/weight/configuration of the module, in consideration of the field layout of the lifting equipment? |
| | | • Have you properly planned the construction work of the junction joining method between modules? | • Agreed | • Have you properly planned the construction work of the junction joining method between modules? |
| | | • Have you simulated the configuration of all junctions between modules in advance? | • Agreed | • Have you simulated the configuration of all junctions between modules with a 3D model in advance? |
| | | - | • (Addition) It is necessary to consider the construction error of the junction. | • Have you considered the construction error of the junction? |
| | | - | • (Addition) It is necessary to consider the workers' accessibility to the junction point. | • Have you reviewed the workers' accessibility to the junction point? |
| Production safety | Consideration of risk factors | • Have you presented an open-storage method by reviewing possible safety issues during open-storage work? | • Agreed | • Have you presented an open-storage method by reviewing possible safety issues during open-storage work? |
| | | • Have you established a lifting plan by reviewing possible safety issues during lifting? | • Agreed | • Have you established a lifting plan by reviewing possible safety issues during lifting? |
| | | • Have you established a joining plan by examining possible safety issues during the joining process? | • Agreed | • Have you established a joining plan by examining possible safety issues during the joining process? |
| Production quality | Minimization of parts breakage | • Not applicable | • (Addition) It is necessary to consider the possibility of deformation, cracks, and partial breakage of the module during on-site work (including open storage, lifting, and joining). | • Have you reviewed the possibility of deformation, cracks, and partial breakage of the module during on-site work (including open storage, lifting, and joining)? |
| | Securement of the quality of junction parts | • Have you selected the joining method in consideration of the junction spacing and the level of stress? | • Agreed | • Have you selected the joining method in consideration of the junction spacing and the level of stress? |
| | | • Have you reviewed the structural performance of the module junction? | • Agreed | • Have you reviewed the structural performance of the module junction? |
| | | • Have you reviewed the use performance (including watertightness, fire resistance, durability, insulation, and sound insulation) of the module junction? | • Agreed | • Have you reviewed the use performance (including watertightness, fire resistance, durability, insulation, and sound insulation) of the module junction? |

**Table 6.** *Cont.*

| Goals of Optimal OSC Design | DfMA Principles | Primary Set of Preliminary Items | Expert Opinions | Secondary Set of Preliminary Items |
|---|---|---|---|---|
| | | • Have you reviewed the ease of vertical/horizontal adjustment of the joining method between modules? (Is it easy to adjust?) | • Agreed | • Have you reviewed the ease of vertical/horizontal adjustment of the joining method between modules? (Is it easy to adjust?) |
| | | • Have you applied a joining material (such as grouting material or hardware) that can ensure good performance against chemical and physical influences? | • Agreed | • Have you applied a joining material (such as grouting material or hardware) that can ensure good performance against chemical and physical influences? |
| | Minimization of the number of parts | • Have you reviewed the possibility of minimizing the number of modules? | • Agreed | • Have you reviewed the possibility of minimizing the number of modules? |
| | Standardization of parts | • Have you minimized the number of module types? | • Agreed | • Have you minimized the number of module types? |
| | | • Have you minimized the number of types of joining methods? | • Agreed | • Have you minimized the number of types of joining methods? |
| | Consideration of the reusability of parts | • Not applicable | • No further comments | • Not applicable |
| Production efficiency | Minimization of additional work | • Have you minimized curing management and special finishing work at the junction point? | • Agreed | • Have you minimized curing management and special finishing work at the junction point? |
| | | • Have you properly configured the size and weight of the module? | • Agreed | • Have you properly configured the size and weight of the module? |
| | Simplification of assembly and handling methods | • Have you minimized the number of junction points of the module? | • Agreed | • Have you minimized the number of junction points of the module? |
| | | • Have you reviewed the difficulty of the joining method? | • Agreed | • Have you reviewed the difficulty of the joining method? |

**Table 7.** The secondary set of preliminary items for the operation & maintenance stage.

| Goals of Optimal OSC Design | DfMA Principles | Primary Set of Preliminary Items | Expert Opinions | Secondary Set of Preliminary Items |
|---|---|---|---|---|
| Production availability | Consideration of manufacturing conditions | • Have you reviewed the location of periodic maintenance activities (including electricity, firefighting, gas, water supply, and rescue) conducted during the building-use phase to minimize user inconvenience? | • Agreed | • Have you reviewed the location of periodic maintenance activities (including electricity, firefighting, gas, water supply, and rescue) conducted during the building-use phase to minimize user inconvenience? |
| Production safety | Consideration of risk factors | • Not applicable | • No further comments | • Not applicable |

**Table 7.** *Cont.*

| Goals of Optimal OSC Design | DfMA Principles | Primary Set of Preliminary Items | Expert Opinions | Secondary Set of Preliminary Items |
|---|---|---|---|---|
| Production quality | Minimization of parts breakage | • Not applicable | • (Addition) Have you considered increasing the durability of modules and junctions? | • Have you considered increasing the durability of modules and junctions? |
| | Securement of the quality of junction parts | • Have you checked and repaired defects (including cracks and leaks) of the junction during the building-use phase? | • Agreed | • Have you checked and repaired defects (including cracks and leaks) of the junction during the building-use phase? |
| Production efficiency | Minimization of the number of parts | • Not applicable | • No further comments | • Not applicable |
| | Standardization of parts | • Not applicable | • No further comments | • Not applicable |
| | Consideration of the reusability of parts | • Not applicable | • No further comments | • Not applicable |
| | Minimization of additional work | • Not applicable | • No further comments | • Not applicable |
| | Simplification of assembly and handling methods | • Is it easy to observe whether a module may have defects, and to access it for maintenance and repair? | • Agreed | • Is it easy to observe whether a module may have defects, and to access it for maintenance and repair? |
| | | • Have you reviewed the difficulty of maintenance and repair of the junction? | • Agreed | • Have you reviewed the difficulty of maintenance and repair of the junction? |
| | | • Have you reviewed the ease of detachment of non-structural partition walls, to improve the possibility of remodeling? | • Agreed | • Have you reviewed the ease of detachment of non-structural partition walls, to improve the possibility of remodeling? |

### *4.3. Derivation of the Final OSC–DfMA Checklist*

This study conducted structured interviews with experts who had experience of participating in PC-based OSC projects, to review its content validity. The expert interviews were conducted in a face-to-face manner from 1 September to 30 November 2021, targeting a total of six experts (two architectural design experts, two structural design experts, one PC-manufacturing expert, and one construction expert). Table 8 shows the demographic characteristics of the experts who participated in interviews.

**Table 8.** Demographic characteristics of experts who participated in content validity evaluation for evaluation items.

| Category | | Frequency | % |
|---|---|---|---|
| Organization type | Architectural Design | 2 | 33.33% |
| | Structural Design | 2 | 33.33% |
| | PC Manufacturing | 1 | 16.67% |
| | Construction | 1 | 16.67% |
| Years of experience | Construction-related | 19.1 years | |
| | OSC-related | 3.8 years (2.9 times) | |
| Total | | 6 | 100% |

The interviews were conducted to evaluate the content validity of the secondary set of preliminary OSC–DfMA items on a 4-point scale (1 [not relevant]; 2 [somewhat relevant]; 3 [quite relevant]; 4 [highly relevant]), as shown in Table 9 [35]. The CVI was calculated using Equation (1), after applying the results evaluated by the six experts for

each evaluation item. The CVI calculation results for each evaluation item are shown in the following Tables 10–13, and only items with a CVI exceeding 0.75 were selected as final OSC–DfMA checklists.

$$CVI = \frac{\text{Number of experts who selected 3 or 4 points}}{\text{Total number of experts who participated evaluation}} \tag{1}$$

**Table 9.** Content validity evaluation criteria for OSC–DfMA items.

| Score | Detail |
|---|---|
| 1 point | Not relevant |
| 2 points | Somewhat relevant |
| 3 points | Quite relevant |
| 4 points | Highly relevant |

**Table 10.** OSC–DfMA items for the manufacturing stage.

| Category | No. | | OSC–DfMA Evaluation Items | CVI | Selected/Not Selected |
|---|---|---|---|---|---|
| Production availability | A1 | • | Have you reviewed the size/weight/shape/configuration of the module in consideration of the manufacturing facilities (such as lifting equipment) at the factory? | 1 | Selected |
| | A2 | • | Was the tolerance for factory production presented? | 0.83 | Selected |
| Production safety | A3 | • | Have you reviewed possible safety issues in the production process of the factory? | 1 | Selected |
| | A4 | • | Have you reviewed the location of the lifting point and the balance of the module during lifting? | 1 | Selected |
| Production quality | A5 | • | Have you reviewed the curing measures for the PC module? | 0.92 | Selected |
| | A6 | • | Have you reviewed the structural performance (including stress and deformation) of the lifting point designated for each module? | 1 | Selected |
| | A7 | • | Have you provided the loading conditions (type, position, and spacing of the pedestal)? | 0.83 | Selected |
| | A8 | • | Have you reviewed the structural performance of the fixing device (anchor bolt) installed at the lifting point? | 0.92 | Selected |
| | A9 | • | Have you reviewed the possibility of module deformation, cracks, and partial breakage during the factory manufacturing process? | 1 | Selected |
| | A10 | • | Have you reviewed the performance (position and route) of the piping connection for water supply, drainage, and sanitation facilities? | 0.83 | Selected |
| | A11 | • | Have you designed considering the application of an appropriate ventilation system? | 0.83 | Selected |
| | A12 | • | Have you reviewed the performance of plumbing connections for electrical and telecommunication facilities? | 0.83 | Selected |
| Production efficiency | A13 | • | Have you reviewed the possibility of minimizing the number of modules? | 1 | Selected |
| | A14 | • | Have you minimized the number of mould types? | 1 | Selected |
| | A15 | • | Have you minimized the number of module types? | 1 | Selected |
| | A16 | • | Have you designed considering the reusability of the mould? | 1 | Selected |
| | A17 | • | Have you properly configured the size and weight of the module? | 1 | Selected |
| | A18 | • | Have you designed considering the difficulty of manufacturing the module? | 1 | Selected |

**Table 11.** OSC–DfMA items for the transport stage.

| Category | No. | OSC–DfMA Evaluation Items | CVI | Selected /Not Selected |
|---|---|---|---|---|
| Production availability | B1 | • Have you reviewed the size/weight/configuration of the module in consideration of transportation equipment? | 1 | Selected |
| | B2 | • Have you reviewed the size/weight/configuration of the module in consideration of road conditions inside and outside the site? | 1 | Selected |
| | B3 | • Have you chosen the size/weight/configuration of the module in consideration of the Road Traffic Act? | 1 | Selected |
| Production safety | B4 | • Have you reviewed possible safety issues during the transport process and presented a transport method that can ensure stable transport? | 1 | Selected |
| Production quality | B5 | • Have you reviewed the possibility of module deformation, cracks, and partial breakage during the transport process? | 1 | Selected |
| Production efficiency | B6 | • Have you reviewed the possibility of minimizing the number of modules? | 1 | Selected |
| | B7 | • Have you minimized the number of module types? | 1 | Selected |
| | B8 | • Have you minimized the use of special transport equipment? | 0.83 | Selected |
| | B9 | • Have you properly configured the size and weight of the module? | 1 | Selected |
| | B10 | • Have you minimized the number of types of transport equipment required to transport from the factory to the site? | 0.83 | Selected |

**Table 12.** OSC–DfMA evaluation items for the on-site assembly stage.

| Category | No. | OSC–DfMA Evaluation Items | CVI | Selected /Not Selected |
|---|---|---|---|---|
| Production availability | C1 | • Have you considered the size/weight/configuration of the module in consideration of the lifting equipment on the site? | 1 | Selected |
| | C2 | • Have you reviewed the size/weight/configuration of the module in consideration of the field layout of the lifting equipment? | 1 | Selected |
| | C3 | • Have you properly planned the construction work of the junction joining method between modules? | 0.92 | Selected |
| | C4 | • Have you simulated the configuration of all junctions between modules in advance? | 0.83 | Selected |
| | C5 | • Have you considered the construction error of the junction? | 0.83 | Selected |
| | C6 | • Have you reviewed the workers' accessibility to the junction point? | 0.92 | Selected |
| Production safety | C7 | • Have you presented an open-storage method, by reviewing possible safety issues during open-storage work? | 1 | Selected |
| | C8 | • Have you established a lifting plan, by reviewing possible safety issues during lifting? | 1 | Selected |
| | C9 | • Have you established a joining plan, by examining possible safety issues during the joining process? | 1 | Selected |

**Table 12.** *Cont.*

| Category | No. | OSC–DfMA Evaluation Items | CVI | Selected /Not Selected |
|---|---|---|---|---|
| Production quality | C10 | • Have you reviewed the possibility of deformation, cracks, and partial breakage of the module during on-site work (including open storage, lifting, and joining)? | 1 | Selected |
| | C11 | • Have you selected the joining method in consideration of the junction spacing and the level of stress? | 0.92 | Selected |
| | C12 | • Have you reviewed the structural performance of the module junction? | 1 | Selected |
| | C13 | • Have you reviewed the use performance (including watertightness, fire resistance, durability, insulation, and sound insulation) of the module junction? | 1 | Selected |
| | C14 | • Have you reviewed the ease of vertical/horizontal adjustment of the joining method between modules? (Is it easy to adjust?) | 1 | Selected |
| | C15 | • Have you applied a joining material (such as grouting material or hardware) that can ensure good performance against chemical and physical influences? | 1 | Selected |
| Production efficiency | C16 | • Have you reviewed the possibility of minimizing the number of modules? | 1 | Selected |
| | C17 | • Have you minimized the number of module types? | 1 | Selected |
| | C18 | • Have you minimized the number of types of joining methods? | 1 | Selected |
| | C19 | • Have you minimized curing management and special finishing work at the junction point? | 0.92 | Selected |
| | C20 | • Have you properly configured the size and weight of the module? | 1 | Selected |
| | C21 | • Have you minimized the number of junction points of the module? | 1 | Selected |
| | C22 | • Have you reviewed the difficulty of the joining method? | 1 | Selected |

**Table 13.** OSC–DfMA evaluation items for the operation and maintenance stage.

| Category | No. | OSC–DfMA Evaluation Items | CVI | Selected /Not Selected |
|---|---|---|---|---|
| Production availability | D1 | • Have you reviewed the location of periodic maintenance activities (including electricity, firefighting, gas, water supply, and rescue) conducted during the building-use phase to minimize user inconvenience? | 1 | Selected |
| Production quality | D2 | • Have you checked and repaired defects (including cracks and leaks) of the junction during the building-use phase? | 1 | Selected |
| | D3 | • Have you considered increasing the durability of modules and junctions? | 1 | Selected |
| Production efficiency | D4 | • Is it easy to observe whether a module may have defects and access it for maintenance and repair? | 1 | Selected |
| | D5 | • Have you reviewed the difficulty of maintenance and repair of the junction? | 1 | Selected |
| | D6 | • Have you reviewed the ease of detachment of non-structural partition walls to improve the possibility of remodeling? | 0.83 | Selected |

In determining OSC–DfMA items for the factory manufacturing stage, the CVI of all items in the secondary set of preliminary items was calculated to be 0.75 or higher. A total of 18 items were thus selected: two items related to production availability; two items related to production safety; eight items related to production quality; and six items related to production efficiency (see Table 10).

In determining OSC–DfMA items for the transport stage, the CVI of all items of the secondary set of preliminary items was calculated to be 0.75 or higher. Thus, a total of 10 items were selected: three items related to production availability; one item related

to production safety; one item related to production quality; and five items related to production efficiency (see Table 11).

In determining OSC–DfMA items for the on-site assembly stage, the CVI of all items of the secondary set of preliminary items was calculated to be 0.75 or higher. Thus, a total of 22 items were selected: six items related to production availability; three items related to production safety; six items related to production quality; and seven items related to production efficiency (see Table 12).

In determining OSC–DfMA items for the operation and maintenance stage, the CVI of all items of the secondary set of preliminary items was calculated to be 0.75 or higher. Thus, a total of six items were selected: one item related to production availability; two items related to production quality; and three items related to production efficiency (see Table 13).

## 5. Conclusions

This study derived OSC–DfMA items suitable for PC-based OSC projects, using the systematic literature review, structured interview, and content validity analysis methods. This study established a frame for deriving a primary set of OSC–DfMA preliminary items reflecting the OSC optimal design goals, the OSC production process, and DfMA principles, through the literature review related to OSC, and used this frame to derive a primary set of OSC–DfMA preliminary items. Furthermore, this study conducted interviews with experts who had experience of participating in OSC projects, so as to include expert opinions in the DfMA checklist. In addition, for verification of the content validity of the derived checklist, interviews with experts who had experience of participating in domestic OSC projects were conducted, to analyze the content validity of each item. Subsequently, the CVI was calculated, and only items with a CVI exceeding 0.75 were selected as final items. Consequently, 18 items related to the manufacturing stage, 10 items related to the transport stage, 22 items related to the on-site assembly stage, and 6 items related to the operation and maintenance stage were selected.

Previous studies related to DfMA in the construction field focused on presenting the development direction of DfMA-related applications and related technologies applicable to OSC, or on providing conceptual design principles. By contrast, this study clearly identified the detailed considerations (checklist) for optimization of design for PC-based OSC projects, giving it significance in providing a foundation for related studies. The checklist presented in this study can be utilized as a guide for optimal design during the process of OSC design, as well as for the inspection process for the optimality of design plans. The results of this study are expected to reduce the possibility of design errors and design modifications in OSC projects.

In the future, we will conduct a case study, to analyze the application effect of the checklist presented in this study. Furthermore, we intend to develop a method for evaluating the degree of optimization, in terms of the production availability, production safety, production quality, and production efficiency of OSC design plans, based on the checklist presented in this study.

**Author Contributions:** Conceptualization, S.J. and J.Y.; methodology, S.J.; formal analysis, S.J.; investigation, S.J.; resources, J.Y.; writing—original draft preparation, S.J.; writing—review and editing, S.J. and J.Y.; supervision, J.Y.; project administration, J.Y.; funding acquisition, J.Y. All authors have read and agreed to the published version of the manuscript.

**Funding:** This work was supported by the Korea Agency for Infrastructure Technology Advancement (KAIA), grant funded by the Ministry of Land, Infrastructure and Transport (Grant 22ORPS-B158109-03). The present research was conducted by the Research Grant of Kwangwoon University in 2021.

**Institutional Review Board Statement:** Not applicable.

**Informed Consent Statement:** Not applicable.

**Data Availability Statement:** The data presented in this study are available on request from the corresponding author.

**Conflicts of Interest:** The authors declare no conflict of interest.

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
