# Peer review of "Design for Manufacturing and Assembly (DfMA) Checklists for Off-Site Construction (OSC) Projects"

_sustainability, doi:10.3390/su141911988_

Round 1

Reviewer 1 Report

This is a quite interesting research item, on the very crucial issue of determining how to design for manufacturing for off-site construction projects. Based on literature review, the authors formulate a check list for assessing the designability for manufacturing of off-site construction projects, which is validated and narrowed down by interviewing 6 experts in the field.

It would add more credibility to the article if this check list was applied to a case study, if possible, to compare how this would improve the off-site construction manufacture. Nonetheless, if this is not possible for the time being, it would be good to establish it as a future goal of the research.

Apart from that, it is important to justify why only 6 experts were chosen and to state whether they were all Korea-based or not. More details should be given on the interviews methodology; structure, type of questions etc.

On the trivial side, the paragraphs in an article are paragraphs and not chapters, while some proof reading should be made to correct some minor linguistic, syntax and semiotic errors.

Reviewer 2 Report

Dear authors,

Overall the manuscript seems to be good. However it needs some revision with suggestions/opinions listed below.

1. Initially define/brief about Design for Manufacturing and Assembly (DfMA) in the introduction part so that readers van have a view on DfMA.

2. What about other countries apart from Singapore, UK & USA to have guidelines for OSC? Any literature support for it? 

3. Last para of introduction seems to be like thesis writing, try to reframe the content or better remove it, chapters 2, 3, 4. ..... and so on.

4. What is the significance of Table 1 DfMA principles in this study? Try to brief about the same.

5. Section 2.4, second para it is stated that, "Many studies have proved the application of DfMA......" how many such studies the authors referred? Any citation?

6. What is the use of content validity index analysis in this study? In what way it is utilized?

7. I felt the structure of the paper needs to be reframed, it looks continues Tables from 2-16 non stop without any discussions. It requires lots of discussion under each table, simply it is stated as "it presents"........ Moreover try to compress or merge the tables which has significant similarities. 

8. How the expert opinions are arrived as mentioned in tables 7, 8, 9 ....? Any protocols adopted for each category, manufacturing, maintenance, assembling?

9. How the score points are considered in the range of 1 pt, 2 pts, 3pts, 4 pts, any references?

10. References are insufficient, try to add more. 

Round 2

Reviewer 2 Report

All suggestions are addressed.